# Exploring the experiences of people who had a stroke and therapists who managed people with stroke during the COVID-19 pandemic: An exploratory qualitative study

**Ahmad Sahely**[1,2]*, **Shara Kai Ning Hew**[1], **Yik Ka Chan**[1], **Andrew Soundy**[1], **Sheeba Rosewilliam**[1]

**1** School of Sports, Exercise and Rehabilitation Sciences, University of Birmingham, Birmingham, United Kingdom, **2** Faculty of Applied Medical Science, Physiotherapy Department, Jazan University, Jazan, Saudi Arabia

* axs1638@student.bham.ac.uk

## Abstract

### Objective

To explore experiences, needs and rehabilitation priorities of patients who had their stroke and the experiences of therapists managing stroke patients during the COVID-19 pandemic.

### Design

Exploratory qualitative study.

### Setting

Acute, sub-acute and community stroke facilities.

### Subjects

Twenty-two participants. Twelve therapists (all female, mean age 38.5 years) and ten patients (9 female, mean age 51.1 years) who were involved in stroke rehabilitation during the pandemic were interviewed.

### Methods

Individual semi-structured interviews were conducted. Interviews were recorded and transcribed before being analysed using a reflexive thematic analysis approach.

### Results

Four main themes demonstrate the modifications in the care system as a result of COVID-19, impact on the stroke patients at different stage, needs and priorities of stroke rehabilitation, and management strategies that have been used in stroke rehabilitation. Remote rehabilitation and self-management strategies were recommended to deliver care for stroke

**Data Availability Statement:** All relevant data are within the manuscript and its Supporting Information files.

**Funding:** The author(s) received no specific funding for this work.

**Competing interests:** The authors have declared that no competing interests exist.

patients. However, therapists seemed unsatisfied with the quality of care delivered and patients suggested face to face delivery of care with proper personal protection equipment to better address their physical and mental health needs.

## Conclusion

The findings of this study explored the impact of the pandemic on stroke care from the perspective of the patients and therapists and provides suggestions for improved delivery of care in similar situations. Future research is warranted to examine the long-term effects on people who had inadequate post-stroke rehabilitation during covid pandemic and urgent measures taken to reduce the impact the pandemic has had on the physical and mental issues for these patients.

## Introduction

Stroke is a major cause of disability worldwide [1]. In the UK, more than 100,000 incidents of stroke had been reported each year prior to the COVID-19 pandemic [2]. Since the start of the pandemic, the admission rate of stroke cases fell markedly [3–5]. For example, in one London hospital there was one third reduction in the number of admissions; however, those admitted had significantly higher pre-stroke disability and greater severity of stroke at admission [5]. The rate of hospitalisation for TIA between March 23rd and 30th June 2020, was significantly lower (24.44%) along with reduced rates of thrombolysis [27 (11.49%) vs. 46 (16.25%, $p = 0.030$)] compared to the same period in 2019. Fear of COVID-19 infections, lockdown isolation, advice from doctors and media could have reduced the prompt admissions and the more severe patients being admitted [6]. It is possible that these patients who had a stroke during the pandemic suffered due to delayed access and reduced opportunities for rehabilitation.

There has been an increasing effort to understand the impact of the pandemic on the healthcare system for people with stroke; most research investigated the disruption in access and delivery of acute services including paramedical services, admission rates, hyperacute and acute-care, and the strain on the staff and caregivers in various settings [3, 7–10]. Only few qualitative studies have focused on the experiences of therapists during the pandemic from a rehabilitation lens, yet none has included patients' perspectives [11–14]. Findings from these studies done in Spain and the United States, highlighted the psychological impact of the pandemic on therapists as front-line workers [11], their professional and ethical challenges when they were re-deployed into COVID-19 units [14], and their overall experience with providing care to the COVID-19 patients [12, 13].

In the UK and other developed countries, recommendations for best practices of stroke care during the pandemic, including rehabilitation, have been developed based on evidence (mainly quantitative data) published during the pandemic and lack the understanding of lived experiences of people which might help better evaluation and addressing of the practice issues [15–17]. The impact of the delayed presentation of people with stroke, resource redirection, and reduced post stroke rehabilitation services on patients' recovery at different stages are still not clear in the current literature. There are also concerns that there will be severe long-term effects on the people who have had their strokes during the pandemic, worsened by the expected increase of having a stroke as one of the post-COVID -19 complications [18, 19].

As the health services have been resuming their normal services, the backlog of healthcare in England continues to mount, which may take years to clear [20]. There is a key need to understand what patients missed during the pandemic and to identify priorities that ensure services can be tailored in any future disruptions. There is a need to be proactive and identify what patients' rehabilitation needs are in preparation for the huge deluge for rehabilitation [21]. The aim of this study was to gain insight into experiences of people with stroke, therapists' experiences, management strategies, patients' needs and rehabilitation priorities after their first stroke during the pandemic.

## Methods

Ethical approval for this study was granted by the Science, Technology, Engineering and Mathematics (STEM) ethics committee at the University of Birmingham (reference: ERN_21–0775). The interviews were collected between September 2021and January 2022. We followed the Consolidated criteria for Reporting Qualitative Research (COREQ) for methodological guidance [22]. A completed COREQ checklist is attached as Appendix D in S2 File.

### Design

The study used an exploratory qualitative design. Qualitative designs are more suitable to investigate individuals' experiences, beliefs, and opinions [23]. The authors of this study take a critical realist perspective that much of reality happens without an awareness of it. This suggests that an external reality exists, but one must access it through an individual's personal world [24]. It aims to produce a deep and thick account of the processes that underpin the effects of pandemic on experiences of people with stroke, their lives and recovery [24].

### Participants

Potential participants were recruited from the community via the websites Stroke Association, Different Strokes, and social media to make contact with stroke patients from diverse geographical and ethnic backgrounds. In addition, therapists were recruited through online professional forums such as the Chartered Society of Physiotherapy (CSP) and the Association of Chartered Physiotherapists Interested in Neurology (ACPIN). To identify a wider range of perspectives, a purposive sampling approach was used to recruit people involved in stroke rehabilitation (patients and therapists) since March 2020 [25]. Therapists were recruited from various stroke clinical settings including hyper-acute, acute, sub-acute and community services to gain a wider insight into their experiences with stroke rehabilitation. At the time of the study many of them were redeployed in other areas due to Covid pressures. However, patient participants were mainly in sub-acute or chronic stage of recovery and they had been discharged into the community; access to patients in the acute setting was not possible because of the Covid restrictions. Eligible participants who contacted the research team to take part in the study were provided with a copy of the study information sheet and asked to sign a written consent electronically before the interviews. Snowball sampling was also employed as participants were asked to pass on the study information to people who they might know who fit the eligibility criteria.

Inclusion criteria for stroke patients were (i) Experienced a stroke (of any type) since March 2020; (ii) age ≥18 years old; (iii) cognitively unimpaired (screened with the Mini Mental State Examination (MMSE) [26] and able to communicate clearly to express their views and provide informed consent; (iv) had access to remote meeting technology such as zoom or phone with speaker technology. The study excluded patients who had a TIA and have less need for services, people with severe cognitive or speech problems or cannot meet remotely

for the interview. For therapists, the inclusion criteria included physiotherapists and occupational therapists who treated people with stroke during the pandemic and can meet remotely for their interviews.

## Data collection

Data collection was done using individual semi-structured interviews using two separate topic guides for therapists and the people who had a stroke during the pandemic. The topic guides explored experiences, opinions, feelings and needs of participants with who had a stroke during the COVID-19 pandemic (attached as Appendices A and B in S2 File). The interview guides were developed by experienced qualitative researchers based on the research objectives. The demographic data that included type of stroke, age, gender, time since stroke, and questionnaires for functional ability using Barthel Index [27], stroke specific quality of life measure [28], mini-mental status evaluation [26], and geriatric depression scale [29] were collected verbally with the patient participants. For therapists, data on age, gender, years of clinical experience, band, level of education, and practice setting were collected.

The data collection process was piloted in a sample of one patient and a therapist, carried out over zoom and recorded for transcription. The researcher made field notes following each interview to record emotions and any relevant situations. Interviews lasted on average 1 hour, ranging between 35 minutes and 1 hour 25 minutes. The primary researcher AS with previous experience in conducting qualitative research collected 80% of data and the other 20% were collected by two assistant researchers under the supervision of AS (YC and SH). The researchers had no prior knowledge of or contact with any participants in the study.

Data saturation guided the sample size for this study [30, 31]. Recruitment stopped with simultaneous data analysis when researchers thought no new themes were emerging and which occurred after we completed 10 interviews with therapists and 8 patient interviews. However, two additional interviews were collected from each group to ensure data saturation was reached.

## Data analysis

Data from interviews were transcribed using a transcription service. Then data was anonymised by removing any identifiable information and was analysed following the reflexive thematic analysis approach [32]. Coding and categorisation of data were done by the main researcher (AS) in addition to two research students (YC and SH) each of who analysed (40%) of data from stroke survivors or therapists independently using iterative analytical methods [33]. The developed codes were shared and discussed between the analysts and then checked by two other senior researchers (SR and AAS) with expertise in qualitative research to improve rigour of data analysis and to solve any disagreement between the analysts. To eliminate researcher's interpretation bias and to increase the integrity of the study's findings, triangulation of codes from different data set (therapists vs. stroke survivors) and between the three researchers was applied [34, 35]. The intention of research team was to use the themes derived from data related to the research objectives to build theory and inform future work.

## Results

### Description of participants

This study included 12 therapists and 10 stroke survivors with a wide range of stroke severity, in different stages of recovery, and from clinical setting and levels of expertise. There were two patient participants who were excluded because of the time after stroke (more than 6 months)

and a therapist who had left the stroke team before the pandemic. Tables 1 and 2 summarise the demographic information of the participants.

**Themes.** The main themes derived from the data demonstrate the impact of modifications in stroke services due to the pandemic on the stroke care system, identify peoples' needs and priorities for stroke rehabilitation at different stages, and provide an insight into the management strategies that were used by different stakeholders in stroke rehabilitation. Fig 1 presents the themes and subthemes that were developed from participants' responses.

*Theme 1*: *Modifications to the services.* Stroke services, including access and delivery of care in the acute and community settings were disrupted. Changes to policy and staffing issues contributed to the disruption further which will be discussed in this theme.

*a. Reluctance to access services.* At the beginning of the pandemic, patients were reluctant to come into hospitals and they suggested that media reports made people to be terrified.

"*the last thing I wanted to do, I've not had COVID all the way through, and I was like we're in March, I haven't had it and if I if I'm going to hospital I'm probably going to get it in hospital*" (Stroke Survivor 1)

In some cases, it was reported that patients did not like to go to hospital unless they felt they had something serious because of their fear of infection. Moreover, a therapist said some cases had not come into hospital immediately for a first stroke.

"*Nobody was presenting themselves to A&E. And then now in retrospect, people have come through the system and had their second stroke, so the first one maybe was a bit more mild and they just cracked on with it.*" (Therapist 1)

Further, some patients mentioned they had misinterpreted stroke presentation to be vaccine related symptoms and as a result, missed the FAST pathway to manage their initial stroke.

"*I had some symptoms. . .but I've put it down to my vaccine because I'd have a vaccine earlier that week.*" (Stroke survivor 1) "*I believe the reason why I had the stroke is because of the Oxford vaccine.*" (Stroke survivor 2)

After discharge, some patients had concerns about home visits because of their fear of infection. Hence it was not surprising when therapists reported a reduction in the admissions to stroke unit along with reduced referrals to community services.

*b. Disruption to the services.* Patients reported that due to lack of direct access to the GPs, which was limited to phone calls or through the NHS app during lockdown, patients with minor symptoms went unnoticed. Patients reported there were delays in their transfer to the hospital and admission to the hyper acute units.

**Table 1. Characteristics of participating therapists.**

| | |
|---|---|
| **Age** | Mean (range) = 38.5 years (24–59) |
| **Gender** | All female |
| **PT/OT** | 4/8 |
| **Years of clinical experience** | Mean (range) = 14.6 years (3–37) |
| **Level of education** | 2 College Diploma/5 BSc/5 MSc |
| **Band** | B5 (1 participant), B6 (4 participants), B7 (6 participants) and B8a (1 participant) |
| **Clinical setting** | 1 hyper-acute, 2 acute, 4 acute and sub-acute, 5 Community |

**Table 2. Characteristics of the participant stroke survivors.**

| | |
|---|---|
| Gender (Male/Female) | 1/9 |
| Age | Mean (range) = 51.1 years (34–79) |
| Time since stroke (months) | Mean (range) = 7.6 months (3–19) |
| Type of stroke (Ischemic/Haemorrhagic) | 8/2 |
| Marital status (single/has partner) | 4/6 |
| The Barthel Index (0–100)* | Mean (range) = 91 (70–100) |
| Stroke Specific Quality of Life (49–245)** | Mean (range) = 125.9 (82–186) |
| Mini Mental Scale Examination (0–30)*** | Mean (range) = 26.6 (23–30) |
| Geriatric Depression Scale (0–15)**** | Mean (range) = 5.8 (12–0) |

* Scores <62.90 indicate moderate disability and scores <21.30 indicate severe disability [36].

** Higher scores indicate better functioning [28]; no indicative score ranges for low, moderate, high were available.

*** A score of 23 or less can be a cut-off point indicating the presence of cognitive impairment [26].

**** A score of > 5 points is suggestive of depression [29].

*" So, I went with them [ambulance] for like, seven, eight hours. And finally, while at the end, they did a brain scan."* (Stroke survivor 3)

Participants indicated that they were unsatisfied with the NHS care due to the issues such as staff shortage, and delay in the diagnostic services such as CT scan. They believe that these challenges had existed prior to the pandemic but, had become worse during the pandemic.

*"I don't think that had to do with the pandemic, it's just the NHS"* (Stroke survivor 3).

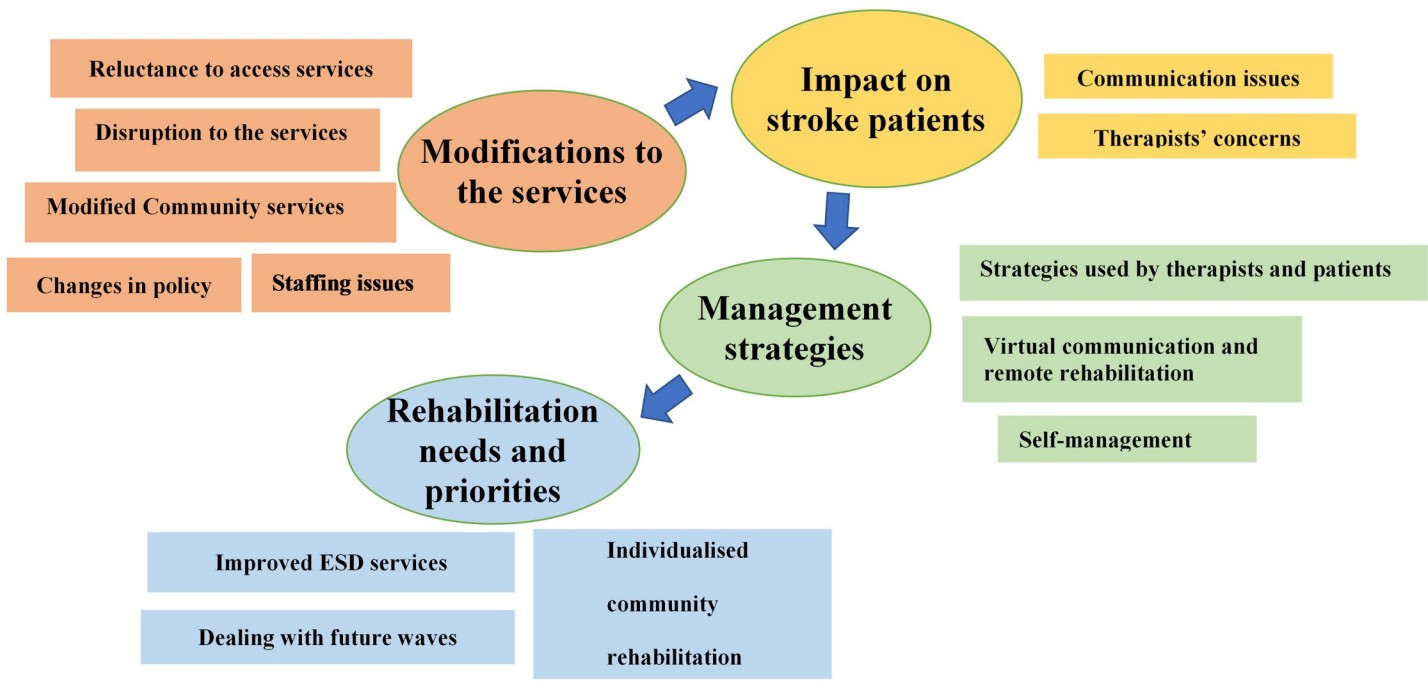

**Fig 1. Themes and subthemes derived from study's data.**

The stroke wards were converted to COVID units which affected the availability of beds for stroke patients and in some cases people were moved to other locations several times. This issue also caused a lack of the therapy equipment and the space required for therapy.

*"No, no. I had nothing in the [acute hospital]. Nothing." (Stroke survivor 2)*

*"So, on our unit we had one of our bays was turned into a unit for COVID patients, so we had a mix of patients, either with or without COVID." (Therapist 2)*

**c. Modified community services.** Some therapists reported that they were under pressure to discharge stroke patients earlier, since COVID patients were a priority for hospital beds. Therapists realised that some patients and their families were not prepared for discharge with no time for preparation or transition with limited handover. Also, there was no access to the gym and other rehab facilities and all therapy groups had to stop. Also, care homes were not accepting patients from the community stroke team.

*"Relatives saying you can't send them home" (Therapist 1) "When people were doing home visits as well, and people couldn't go home to do a home visit to see if there was any equipment required, so again that was like some of the discharges weren't as safe as they should be."(Therapist 5)*

Concurrently, there was a reduction in community services for patients after discharge from hospital.

*"So, they didn't come. I was discharged in December and the first time they came was in April this year." (Stroke Survivor 5)*

Patients also reported lack of therapists' commitment to home visits and delay in delivering essential equipment and services. Patients reported their feeling of being left alone after moving into the community and there was a lack of clear rehabilitation plans.

*"No one rang me, they came and assessed me one day, and then they told me, we're sorry we won't be able to send physio. . . Well, I cried when they left, how can I wait 12 weeks for physio. So, I started then thinking, I might as well be dead. I may as well not recover because I can't just sit here and be like this." (Stroke Survivor 7)*

*"I feel like at the time of discharge, I was just left to it." (Stroke Survivor 2). "But you would have sent me home alone, which I think would have been a massive struggle for me, I could see." (Stroke Survivor 3)*

Contrary to the above in some cases, Early Supported Discharge services did not stop (face to face visits) for newly discharged patients. However, therapists were concerned about the limited hours for stroke rehabilitation and that patients were given little if reached in the community (only an exercise sheet on request) in some cases.

During the lookdowns, patients became very deconditioned as they were not allowed to go into shops to do mobility and shopping practice and lacked social interaction with people which increased risk of mental and emotional problems.

*"We saw a lot of people coming into hospital, having become very deconditioned over the lockdown period because people stopped going out" (Therapist 4) "They could have been sat in the care home for 4–5 months without any input, . . .. So, they were put at a disadvantage, and it's*

*quite a lot of them patients, that when we eventually got to them, that they've been that decon- ditioned it's deemed that you know this is their new baseline, they don't actually have any rehab potential." (Therapist 2) "patients were very isolated, very bored, very depressed, very scared. Physically they went down as well, downhill" (Therapist 6)*

**d. Changes in policy.** Therapists reported that as a part of safety assurance for people in the care facilities, risk assessments were introduced for therapists and patients. Therapists were classified into (clean & dirty/hot & cold) and therapists with high risk of COVID cannot see patients. Triage systems such as the traffic light (Red, Amber, Green) rating system were used to prioritise patients for receiving care. The triage and initial assessments were done mostly by phone for patients immediately after they were discharged.

*"We kind of put a system in place, which was "red, amber, green" . . . . . . So, if they were green, we could go straight in once they were discharged from hospital. Amber the vulnerable, and red would be even if they are showing signs of COVID. . . . . . it works really well, because it helped us identify what patients definitely needed with the face-to-face intervention." (Thera- pist 4)*

Checking against COVID-19 symptoms, wearing of the Personal Protective Equipment (PPE), and avoiding the mix of COVID-19 infected patients with non-infected patients were some of the new policies implemented in therapists' daily routine. In some contexts, there were some additional measures that were added based on personal judgments such as wearing fresh uniforms and washing shoes daily. Therapists were frustrated by these measures espe- cially since proper PPE was not provided at some trusts and COVID-19 test were overwhelm- ing as they took a long time to implement at the beginning of the pandemic.

**e. Staffing issues.** Staff deployment was happening on different levels. For instance, stroke therapists and nurses were deployed to work on COVID-19 wards. There was also a deploy- ment of staff from other teams to make sure that there's enough staff on the stroke ward.

*"And they repatriated quite a lot of staff from the community, they pulled them into the acute setting. . . . . .there weren't enough staff sitting in the community." (Therapist 2)*

This has been considered by some participants as taking a new role without proper training and raised a concern of uncertainty about what is coming.

*"We just became a generic therapist. we weren't physios or OTs or speech therapist. So, when the nurses were really short because half of them had COVID, you ended up doing your ther- apy quite functionally. So, you'd be helping to feed them at lunchtime or walking them to the toilet, or helping them wash, things that as a physio I wouldn't usually do." (Therapist 7)*

*Theme 2*: *Impact on stroke patients*. Patients felt quite confused, because everything just happened so quickly. Some of them reported their frustration over poor care and losing the chance for better recovery as a result of the disruption to the services. Because of the staffing issue (i.e., staff shortage and deployment), some patients thought they had an awful experience under the care of unqualified nurses during the pandemic and received inadequate rehabilita- tion in the hospital.

*"I don't think they could take care of me, though, there wasn't enough staff (Stroke survivor 6) "I do feel sorry for the nursing staff because so we're under an awful pressure, awful lot of*

*pressure, and I know I'm complaining, I'm not complaining but giving it to you as it is, and I don't like to put the NHS apart, they do a sterling job, but my experience, because of the pandemic, I think this horrendous. And I never ever want to go through it again." (Stroke survivor 2)*

Therapists and patient participants had similar concerns that patients' needs for rehabilitation were not met and that they were susceptible to physical and mental health complications.

Therapists suggested that complications due to lack of therapy during acute stages will increase pressure on health care services due to future complications.

Stroke patients who got COVID had to stay longer in hospital. Yet, the infected patients had delayed rehabilitation.

*"And then I caught COVID in the hospital. I was also doubly infected. I think it delayed my beginning of physio and everything." (Stroke Survivor 3)*

**a. Communication issues.** Inpatients experienced difficulty in communication with staff, because of the PPE and social distancing measures. This led to a lack of facial recognition and made it hard to build a healthy therapeutic relationship with inpatients. Inpatients were socially isolated as they had no communication with staff or other patients on the ward. Also, interpreters were not allowed to come to help non-English speakers.

*"I think those patients for who perhaps English isn't their first language it was really tricky, because initially it was difficult to get formal interpreters to come to the ward." (Therapist 8)*

Patients expressed devastation due to lack of family support and therapists reported losing the support of family in caring for inpatients. There was a feeling of loneliness and lack of family/friends support as no visitors were allowed.

*"If the hospital could have let my daughter come in, or my granddaughter, or my friends, they could have done a lot for me while they were visiting. But as it was, everything I wanted, every time I wanted to go to the toilet, I had to call the nurse." (Stroke Survivor 4). "I didn't see any family for three and a half months; it was a long time." (Stroke Survivor 3). "This was one of their big issues at that time, not having a proper communication with their families and friends, and kind of feeling isolated at that time." (Therapist 5).*

In some cases, the lack of mental and emotional support during the inpatient period had led patients to ask for early discharge from hospital.

*"The sister on my ward. . .hasn't got time to talk, I never had a conversation with her ever. . . There was nothing said about your mental health, we were just left." (Stroke Survivor 4)*

Additionally, the lack of access to the internet, TV or any other entertainments made it worse while staying in the hospital.

**b. Therapists' concerns.** Some therapist participants expressed sadness over losing some of their colleagues in the pandemic and that had made them more scared of the situation. Therapists expressed their frustration with the management as they thought they should have had more training opportunities for staff and teams in dealing with similar situations. They indicated that it was an even more challenging time for newly qualified therapists.

*"Some are in their third year of placement. . .. . .the amount of hands on contact they've had with a patient is very minimal. So that's going to have a huge implication on how quickly we going to get out to patients, and also it's going to take us a lot more time trying to train up the band fives confidence." (Therapist 3)*

With regard to patients, therapists reported their feeling of letting patients down as no quality services were provided along the rehab journey even in the acute settings. While some therapists were keen to provide help for their patients, they had concerns about safety and infection control, which limited their support to patients.

*"It is a difficult balance" (Therapist 1) "You couldn't progress them over the phone onto a walking stick, you just literally couldn't do that, you'd have to do that face to face with them." (Therapist 6)*

*Theme 3*: *Management strategies*. This theme summaries the strategies that have been used by service providers or stroke survivors and their carers to manage the rehabilitation process and deal with challenges. Some strategies were implemented based on organisational and clinical practice guidelines such as early supported discharge and remote rehabilitation and others were initiated by patients such as self-management, goal setting and seeking social support.

**a. Strategies used by therapists and patients.** In the inpatient settings, some participants managed to virtually communicate with relatives and friends. In some hospitals, this was facilitated by care teams to enable communication of inpatients with their families.

In the community, patients had to manage their own rehabilitation at home as no services were provided.

*"I was already independent before they [community physiotherapists] came in! What I did, I exercised myself, and I was determined to get downstairs. By the time physios came 12 weeks later, I was already doing the stairs and getting down to the bottom." (Stroke Survivor 8)*

Some participants said they felt the need for private care while waiting for community services. However, private care was costly and participants were unsatisfied with their care quality. Both groups of participants liked how YouTube videos, online games, online stroke groups, social media helped their rehabilitation, especially when patients had financial challenges to pay for private services.

*"There are games which the patient can do, so that can almost be linked to a therapist device, so they could see how well they are getting on and how much rehab they're doing with their hand each day, you can set targets. . .. . . it tracks their progress and how much rehab they have done on it.". (Therapist 9)*

Some therapists also suggested photographing of the exercise programmes given in the hospital was useful for easier home implementation. Also, lending iPads to patients was a strategy used by some teams to enable patients to access rehabilitation remotely.

**b. Virtual communication and remote rehabilitation.** Therapists indicated that the biggest modification for community teams was replacing face-to-face rehabilitation in the community with virtual communication. Virtual communication emerged as a new strategy to reduce direct contact with patients; this included virtual patients' assessments, remote delivery of therapy and follow up. Remote rehabilitation was implemented as the only option for the delivery of rehabilitation therapy for patients post discharge. Some therapists were not happy with the

new policies regarding virtual rehabilitation at the beginning because of the lack of technology in the department to deliver services, difficulty in communication with some stroke survivors, and virtual assessments were not always accurate. It was also challenging for some patients who did not have a device or any access to the internet, not familiar with technology, without carer support and were worried about their safety at home. Some trusts took some time to figure out a platform for virtual delivery of services, which raised staff frustration.

*"There people who have refused onward referral, because they don't want people in their home, and they haven't got access to technology, or they are not in a position to use that."* *(Therapist 8)*

However, remote rehabilitation was appreciated by both groups as it was the only way of providing rehabilitation therapy. Therapists also alluded to video-communication as more effective than Tele-care as it allows them to see patients. They also suggested that younger stroke survivors were happy with online support, accepted and benefitted more from the remote therapy. Some therapists viewed virtual communication positively and thought it was more efficient (giving them time to contact more patients per day without physical travelling). They felt phone assessments were quicker and enabled collection of more detailed information.

**c. Self-management (SM) strategies.** Although some therapists believe that SM should always be delivered as an important part of rehabilitation even before COVID, most therapists indicated their increasing effort to empower patients and their families for self-management during the pandemic. Patients themselves demonstrated self-motivation to carry out SM without any support from the NHS. There were times where SM was the only option to improve independence after discharge. Further, patients suggested that goal setting was vital for their rehabilitation as it pushed the boundaries for the individuals.

*"They [the process of goal setting] are vital, absolutely vital, because I know what I'm working on, I've got direction." (Stroke Survivor 8)*

Some Early supported discharge (ESD) teams encouraged referrers to prepare patients for self-management and home exercises before they were referred to them. ESD therapists had also facilitated patients' discharge by providing education, involving patients in goal setting, and planning for training for the various stages of recovery. The constraint induced movement therapy, stretch classes and GRASP were mentioned as good examples of SM programmes.

*"There was a big move towards giving more education whilst they are in inpatient, so that they were more informed about things like signs, symptoms, FAST, and perhaps more focus in terms of medication, again who to liaise with. . . . . . trying to empower someone, whether they were able to more self-manage." (Therapist 10)* Another therapist explained that as *"we tell patients: We come. We guide. You do.' Instead of we come and do with you" (Therapist 11)*

Providing information packages and compiling apps and online resources from the Stroke Association or Different Strokes were other ways of supporting patients' information needs. Some patients found YouTube and online resources were helpful for teaching SM. Therapists and patient participants agreed that family motivation and encouragement could improve recovery at home.

*"It was my family, my daughter. She did everything she could, she encouraged me to be moti-vated, at the time my mental health was down." (Stroke Survivor 2)*

Contrary to the support for SM, some therapists thought that patients' engagement in self-management was very variable ('self-management is not suitable for everyone'). Therapists suggested that the success of self-management strategies can be affected by fatigue, safety, motivation, self-efficacy and lack of preparation for SM. Also, patients explained difficulty of engaging in SM in the early days post discharge and for people with low functional capacity. They also suggested that an individual's SM plan should be regularly checked by professionals.

*" I think there needs to be more regular check-ins from, perhaps GP or NHS personnel for stroke survivors. Especially from a learning curve. . . . . . So, if you're going down the self-man-agement route, then somebody has to govern and make sure that person's okay to do that independent work." (Stroke survivor 10)*

*Theme 4*: *Rehabilitation needs and priorities.* **a. Improved ESD services.** Therapists believe people who had stroke during the pandemic experienced longer and slower rehabilitation. Therefore, they suggested that the main priority was maintaining the ESD and the community services on discharge.

*" being able to see people on going in their own home, would be the better thing for these patients". (Therapist 2) " I think, probably, maintaining the ESD and the community services on discharge, and how that happens, is probably the most important thing" (Therapist 4)*

They indicated it was important to make a seamless service across the pathway starting with normal inpatient care with proper PPE. To improve inpatient care, therapists suggested better communication and emotional support, family visitations and participation in therapy sessions. Having sufficient staff and training especially for the ESD teams, more funding for social support services, preparing patients for discharge, psychological support and providing better resources for remote rehabilitation with family engagement can help to address rehabilitation needs and manage the waiting list for the community services.

*"During the pandemic we managed, and now we're not managing. We can't get staff, and just so many more referrals, so I think that's the main priority that needs sorting out. People need the rehab straightaway, they need the physio straightaway.". (Therapist 8)*

**b. Individualised community rehabilitation.** Patients suggested engagement in individual-ised rehabilitation plans post discharge from acute care. Mobility was considered by partici-pants as the main goal of physical rehabilitation to improve independence.

*"My first goal was to stand up. . .. And I just said, I want to stand up, because I need stand up so that I can be transferred onto one of the pieces of equipment so I can move to the toilet and so." (Stroke survivor 6)*

Access to the gym and other equipment and joining therapy groups seemed to be a priority for patients if it was suitable for them.

*"I hope there would be more OT [Occupation Therapy], I hope there will be more physio. I really like to be in a gym, I'd like to be able to use equipment to do things." (Stroke survivor 5)*

Patients indicated support for their mental health needs in different stages of their recovery. Participants said more conversations and information was needed for mental health in addition to support from the GPs.

*"I think they should talk to you more, tell you about what's happening, what the prospects are, what you can do to help your mental health, give you more encouragement". (Stroke survivor 6)*

**c. Dealing with future waves.** Some therapists described the current practice as a 'new normal' in which services are meant to go back to pre-COVID19 practice with new measures that can help to improve health and safety in future waves. The 'new normal' also gives the option for virtual communication and remote rehabilitation if needed and this was agreed by patients.

*" I do have a patient currently now who still doesn't really want us to visit. She's much happier for us to converse through WhatsApp calls rather than a face-to-face, and we respect that". (Therapist 12)*

Participants discussed some plans on how things can be better done in the future. For instance, some therapists suggested strategic planning of how to manage their caseload in case of service disruption instead of redeploying staff or a complete stopping of services.

*"I think there have to be lessons learned about trying to continue with community services and not redeploying staff"* (*Therapist 8*)

They indicated that the assessment in the beginning and the triaging are important to prioritise care of most needed individuals. Face to face rehab was suggested (by therapists and patients) in future waves with proper PPE even for patients in isolation.

*"There is no reason for us not to continue normal care for patients". (Therapist 9)*

Therapists suggested remote rehabilitation can be implemented even for individuals who need to be isolated under some circumstances However, safety assurance and additional technology were required for practical delivery. Also, patient education and family involvement could help deliver rehabilitation in the community in addition to the Apps and online groups.

*"It was in small group, people experiencing something like me, so we understand each other so we encourage each other." (Stroke Survivor 3)*

Patients and therapists agreed that relatives should be allowed to visit inpatients. In this case, infection control could be ensured by applying on site testing for COVID that can be more practical.

## Discussion

This study is one of the first to describe the impact of the COVID-19 pandemic on stroke rehabilitation from the stakeholders' perspective in England. The findings of this study describe how the pandemic affected the stroke care system in various settings, modifications in patients' and therapists' practice, strategies that were used to manage the disruption in the services and outline the needs and priorities for future rehabilitation practice. The data was collected from

patients and therapists' interviews with various demographic characteristics thus improving validity and creditability of findings. Including participants from different geographical areas showed that there was variability in the organisational response to the pandemic in various settings. These differences in responses in various settings might have been influenced by the rapid spread of the pandemic locally, the daily updates leading to panic and ambiguous guidance from the local decision makers.

Interestingly, some patient participants mentioned that they thought their stroke happened because of their earlier COVID-19 infection which was also evident in other medical research findings [18, 19]. Moreover, participants indicated that the A&E staff and they were confused between stroke and COVID-19 as a cause of their clinical symptoms. Stroke symptoms like musculoskeletal pain and fatigue, headache, blood pressure, and vision problems were commonly found in patients with Covid. This issue might highlight the need for better differential diagnosis techniques when making a new diagnosis for people with a stroke in this pandemic era.

Participants in our study indicated that restrictions on communication and family visits in the acute settings had negatively affected their recovery. Similarly, participants in another study involving clinicians working in the intensive care units across the UK, showed their dissatisfaction with their way of interacting with patients' families during the pandemic and they expected patients, and families, to be negatively impacted due to this [37]. In some cases, this issue was successfully solved by allowing inpatients to use technology to communicate with staff and see their families/friends remotely. It was even suggested to improve infrastructure for all inpatients facilities with proper technology to allow patients remote social communication that also reduce the need for physical visits as a part of the new normal services.

The findings of this study demonstrate the lack of quality of care along the continuum of stroke care in terms of rehabilitation. Patients expressed their frustration with the impact the pandemic had made on the services and suggested face to face delivery of the rehabilitation services with proper precautions. Though most of the recommendations for best practices during the pandemic, suggest remote communication with patients as an ideal approach [17, 38] patients still wanted face to face delivery and communication. There is a need to prepare and educate patients on the uptake and benefits of remote rehabilitation.

Despite the findings of some studies that delivery of stroke acute care has been maintained or even improved during the lockdown in the UK and other high-income countries [3, 15], our findings show a level of incongruence between system effectiveness from an organisational perspective versus patient satisfaction about services. Also, the previous studies did not discuss community services which have faced more challenges and did not meet standards in some cases. This inadequacy to meet patient needs was reflected in the fact some of the narratives were emotional and patients requested for rest during the interviews. They alluded this emotional upset was due to the lack of psychological support especially during the pandemic and hoped to get that as a priority of care when services are restored.

Despite the relaxing of the COVID-19 measures and coming back to normal service delivery, the most striking issue in current practice for community services is how to manage the long waiting list of patients especially with the current early discharge policy. Our findings support the results of another study that early supported discharge plans for stroke patients were inappropriate, such as being unsafe and uncoordinated [39]. Patients were found to be discharged with delays in installing home equipment and further community rehabilitation. Moreover, families were inadequately prepared to care for stroke patients at home as a result of restrictions on visitations and communication with family during hospitalisation which was also identified as a big challenge for the caregivers of stroke patients in another study in the United States [9]. Caregivers in that study thought that frequent attendance of therapy sessions

and direct communication with staff could improve their readiness for the discharge of their family members from inpatient rehabilitation. As that was impossible during the pandemic, they suggested alternative measures such as scheduled phone calls from all members of the care team and frequent video conference or footage from therapy sessions to keep the caregiver involved in the plan of care [9].

The healthcare needs of caregivers emerged strongly from participants' comments. Some carers and family members tended to display emotional vulnerability for the caring roles due to lack of social or psychological support. In a national stroke survey, 78% of the informal carers had alarmingly presented with severe mental health impacts during the pandemic [40]; psychosocial support needs to be extended to not only the stroke survivors but also their carers of people who had a stroke during the pandemic. In our study, community psychologist consultations received by a participant's family member was suggested as helpful; professional counselling might be one of the resolutions for caregivers to lessen their emotional burden and could in return benefit stroke patients' recovery [41].

The use of remote rehabilitation, online groups, and SM approaches have been found very effective for some patients in previous literature [42, 43]. During the pandemic, dependence on these approaches as the key strategy for improving recovery urges further uptake and adoption of SM during the period when the services are still catching up. However, SM was not the best option for everyone and the capacity of an individual for self-management involving remote rehabilitation should be evaluated [44]. Also, our findings confirm some challenges for the delivery of remote rehabilitation identified by previous research such as users' poor IT skills and resources, miscommunication between individuals, and safety issues [45]. The remote delivery of SM is found to work best in cases of motivated survivors, having a supporting caregiver and when therapists have a healthy relationship with their patients. These factors should be considered when planning for individual care and when planning patients' groups for therapy.

There were some limitations to this study. Some participants were unsatisfied with meeting online for the interviews, especially stroke survivors who weren't familiar with the use of technology. Consequently, the researcher had to do multiple meetings to carry out assessments and interviews in separate sessions. Most of our participants were female in both groups (only one male survivor), which might not represent the perspectives of male stroke survivors in the UK (males' prevalence (51%) > females) [46]. However, it was not feasible for this study to purposefully recruit more participants based on their gender. This study included participants from Birmingham and London areas. Hence, to develop a national recommendation for best practices there might be a need for national research that can evaluate the system more widely. Also, the study only included patients from sub-acute and chronic stages; due to this limitation, we might not be entirely sure of the challenges faced by patients in the acute settings. However, as previously explained, access to acute patients was very restricted during the Covid pandemic. The findings of this study demonstrate the situation of the stroke services during the pandemic, when the interviews were collected, and future research can evaluate the long-term impact of the pandemic on service delivery after the relaxation of the control measures and when services return to normal or the new normal practice. A key question remains as what the residual long-term impact of the pandemic is on the health care system and patients who have had stroke during this pandemic situation.

## Conclusion

The findings of this study demonstrated the impact of the pandemic on stroke care and provided suggestions for optimal delivery of care in future situations. Future research is warranted

to examine the efficacy of new strategies such as reducing the time for acute care and enhancing remote delivery of rehabilitation for people with mild to moderate impairments post stroke.

## Clinical messages

- Despite patients recommending face to face delivery with precautions, when offered self-management interventions, their ability to engage in self-management seemed to be activated since professional support was limited during the lockdown.

- Patients' competence to engage in self-management and remote rehabilitation should be assessed prior to delivering self-management due to challenges related to access and ability to use technology.

- According to patients, the use of technology was highly recommended as the only hope for participating in rehabilitation during the pandemic.

- Future clinical practice should consider the digital divide and improve opportunities for Information Technology education.

## Supporting information

**S1 File.**
(PDF)

**S2 File.**
(DOCX)

## Acknowledgments

The authors would like to acknowledge the patients and therapists who participated. AS, SR, and AAS contributed to the design of the study. AS was responsible for acquisition of data. All authors contributed to analyses and interpretation of data. All authors contributed substantially to drafting the article or revising it critically.

## Author Contributions

**Conceptualization:** Ahmad Sahely, Andrew Soundy, Sheeba Rosewilliam.

**Data curation:** Ahmad Sahely, Andrew Soundy, Sheeba Rosewilliam.

**Formal analysis:** Ahmad Sahely, Shara Kai Ning Hew, Yik Ka Chan, Andrew Soundy, Sheeba Rosewilliam.

**Funding acquisition:** Ahmad Sahely.

**Investigation:** Ahmad Sahely, Shara Kai Ning Hew, Yik Ka Chan, Andrew Soundy, Sheeba Rosewilliam.

**Methodology:** Ahmad Sahely, Andrew Soundy, Sheeba Rosewilliam.

**Project administration:** Ahmad Sahely, Sheeba Rosewilliam.

**Resources:** Ahmad Sahely, Sheeba Rosewilliam.

**Software:** Ahmad Sahely, Sheeba Rosewilliam.

**Supervision:** Ahmad Sahely, Andrew Soundy, Sheeba Rosewilliam.

**Validation:** Ahmad Sahely, Andrew Soundy, Sheeba Rosewilliam.

**Visualization:** Ahmad Sahely, Andrew Soundy, Sheeba Rosewilliam.

**Writing – original draft:** Ahmad Sahely.

**Writing – review & editing:** Ahmad Sahely, Shara Kai Ning Hew, Yik Ka Chan, Andrew Soundy, Sheeba Rosewilliam.

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
