## [Decision Letter · Decision Letter 0]

12 Dec 2022

PONE-D-22-26996Exploring the experiences of people who had a stroke and therapists who managed people with stroke during the Covid-19 pandemic: An exploratory qualitative studyPLOS ONE

Dear Dr. Sahely,

Thank you for submitting your manuscript to PLOS ONE. After careful consideration, we feel that it has merit but does not fully meet PLOS ONE’s publication criteria as it currently stands. Therefore, we invite you to submit a revised version of the manuscript that addresses the points raised during the review process.

Your manuscript has been assessed by one peer-reviewer and their report is appended below.  The reviewer comments that aspects of the manuscript could be strengthened with some additional detail and clarification. Furthermore, the reviewer states that the inclusion of therapist quotes might be beneficial, and the inclusion of a key for table 2 would help the reader interpret the scores.  Could you please revise the manuscript to carefully address the concerns raised? Please note that we have only been able to secure a single reviewer to assess your manuscript. We are issuing a decision on your manuscript at this point to prevent further delays in the evaluation of your manuscript. Please be aware that the editor who handles your revised manuscript might find it necessary to invite additional reviewers to assess this work once the revised manuscript is submitted. However, we will aim to proceed on the basis of this single review if possible. 

We look forward to receiving your revised manuscript.

Kind regards,

Maria Elisabeth Johanna Zalm, Ph.D

Editorial Office

PLOS ONE

Journal Requirements:

Reviewers' comments:

Reviewer's Responses to Questions

**Comments to the Author**

1. Is the manuscript technically sound, and do the data support the conclusions?

Reviewer #1: Yes

2. Has the statistical analysis been performed appropriately and rigorously? 

Reviewer #1: N/A

3. Have the authors made all data underlying the findings in their manuscript fully available?

Reviewer #1: Yes

4. Is the manuscript presented in an intelligible fashion and written in standard English?

Reviewer #1: Yes

5. Review Comments to the Author

Reviewer #1: Very interesting and important topic.

Well written manuscript.

Valuable learning from patients and therapist perspective on the impact of the pandemic on the post stroke journey.

Sound study design and methodology.

Detailed inclusion and exclusion criteria and sampling strategy.

Some suggested minor revisions as follows:

1. The description of the different stages of recovery acute, subacute and community are specified in the abstract and in the results. Make a more explicit link to this point in the sampling and eligibility criteria in the methods.

2. For Table 2 add a key to interpret the scores of the BI, SSQOL, MMSE and GDS?

3. Results - for most of the themes there is a balance between stroke survivor and therapist quotes. For some of the themes eg disruption of services, could the authors include a therapists quote? likewise this is the case for communication issues, a therapist quote could be included here as well?

4. Typo, Discussion para 3, line 1 list should be visits?

5. Recall of acute and subacute stages is this a potential limitation as well?

6. Clinical messages - consider merging the two points relating self management and add a clinical message regarding use of technology, reflecting on the experiences found in this work and future areas to consider in relation to the use of technology in stroke rehabilitation.

7. Include the completed COREQ checklist as an appendix?

6. PLOS authors have the option to publish the peer review history of their article (what does this mean?). If published, this will include your full peer review and any attached files.

Reviewer #1: No

---

## [Author Response · Author response to Decision Letter 0]

27 Jan 2023

Thank you very much for your helpful comments on this manuscript. An effort was made to address the comments and make a proper submission. The revised manuscript was strengthened with some additional details and clarification as suggested and we are looking forwards for publication.

Kind Regards

---

## [Decision Letter · Decision Letter 1]

14 Feb 2023

Exploring the experiences of people who had a stroke and therapists who managed people with stroke during the Covid-19 pandemic: An exploratory qualitative study

PONE-D-22-26996R1

Dear Dr. Sahely,

We’re pleased to inform you that your manuscript has been judged scientifically suitable for publication and will be formally accepted for publication once it meets all outstanding technical requirements.

Kind regards,

Walid Kamal Abdelbasset, Ph.D.

Academic Editor

PLOS ONE

Additional Editor Comments (optional):

Reviewers' comments:

Reviewer's Responses to Questions

**Comments to the Author**

1. If the authors have adequately addressed your comments raised in a previous round of review and you feel that this manuscript is now acceptable for publication, you may indicate that here to bypass the “Comments to the Author” section, enter your conflict of interest statement in the “Confidential to Editor” section, and submit your "Accept" recommendation.

Reviewer #1: All comments have been addressed

Reviewer #2: All comments have been addressed

2. Is the manuscript technically sound, and do the data support the conclusions?

Reviewer #1: Yes

Reviewer #2: Yes

3. Has the statistical analysis been performed appropriately and rigorously? 

Reviewer #1: N/A

Reviewer #2: Yes

4. Have the authors made all data underlying the findings in their manuscript fully available?

Reviewer #1: Yes

Reviewer #2: Yes

5. Is the manuscript presented in an intelligible fashion and written in standard English?

Reviewer #1: Yes

Reviewer #2: Yes

6. Review Comments to the Author

Reviewer #1: (No Response)

Reviewer #2: Comments provided by reviewers addressed

Thank you

7. PLOS authors have the option to publish the peer review history of their article (what does this mean?). If published, this will include your full peer review and any attached files.

Reviewer #1: No

Reviewer #2: **Yes: **Tamer M. Shousha

---

## [Editor Report · Acceptance letter]

17 Feb 2023

PONE-D-22-26996R1 

Exploring the experiences of people who had a stroke and therapists who managed people with stroke during the Covid-19 pandemic: An exploratory qualitative study 

Dear Dr. Sahely:

I'm pleased to inform you that your manuscript has been deemed suitable for publication in PLOS ONE. Congratulations! Your manuscript is now with our production department. 

Kind regards, 

on behalf of

Dr. Walid Kamal Abdelbasset 

Academic Editor

PLOS ONE